# LLMs Can Achieve High-quality Simultaneous Machine Translation as Efficiently as Offline

## Abstract

When the complete source sentence is provided, Large Language Models (LLMs) perform excellently in offline machine translation even with a simple prompt "*Translate the following sentence from [src lang] into [tgt lang]:*". However, in many real scenarios, the source tokens arrive in a streaming manner and simultaneous machine translation (SiMT) is required, then the **efficiency** and **performance** of decoder-only LLMs are significantly limited by their auto-regressive nature. To enable LLMs to achieve high-quality SiMT as efficiently as offline translation, we propose a novel paradigm that includes constructing supervised fine-tuning (SFT) data for SiMT, along with new training and inference strategies. To replicate the token input/output (I/O) stream in SiMT, the source and target tokens are rearranged into an interleaved sequence, separated by special tokens according to varying latency requirements. This enables powerful LLMs to learn read and write operations adaptively, based on varying latency prompts, while still maintaining efficient auto-regressive decoding. Experimental results demonstrate that, even with limited SFT data, our approach achieves state-of-the-art performance across various simultaneous translation benchmarks and different evaluation metrics, and preserves the original capabilities of offline translation. Moreover, EAST generalizes well to document-level SiMT setting without requiring specific fine-tuning, even beyond the offline translation model.

## 1 Introduction

Simultaneous machine translation (SiMT) (Gu et al., 2017; Ma et al., 2019) is a critical technique for enabling seamless cross-linguistic communication in real-time scenarios, such as international conferences. Unlike offline neural machine translation (NMT), where the entire source sentence is available before translation begins, SiMT systems start translating before receiving the complete input, achieving a balance between translation quality and latency.

Large language models (LLMs) have achieved significant advances in NMT, demonstrating impressive capacity when translating full sentences in offline settings (Xu et al., 2024a;b; Guo et al., 2024a; Feng et al., 2024). However, their application to SiMT remains underexplored and faces several significant challenges. First, most existing SiMT models (Zhao et al., 2023; Guo et al., 2024b; Raffel et al., 2024) are typically trained on offline NMT data due to the scarcity of SiMT-specific datasets. This training setup does not align well with the demands of SiMT, which hinders the model's ability to learn how to translate effectively with incomplete input (Wang et al., 2023b; Sakai et al., 2024). Second, many SiMT approaches focus on optimizing prompt structures to simulate SiMT for LLMs (Wang et al., 2023a; Koshkin et al., 2024a;b; Guo et al., 2024b; Agostinelli et al., 2024; Cheng et al., 2024), which typically requires recomputing the key-value (KV) cache since the prompt changes continuously with the update of the source and target. This recomputation significantly increases the computational cost and inference latency, limiting the efficiency of SiMT systems (Raffel et al., 2024). Lastly, LLMs-based methods typically employs fixed read-write policies (Wang et al., 2023a; Agostinelli et al., 2024; Sakai et al., 2024; Wang et al., 2024; Raffel et al., 2024), such as the wait-$k$, to achieve low-latency translations for LLMs. However, these methods fail to adaptively adjust its read/write actions based on sentence structure and context, leading to suboptimal translation quality.

In this paper, we introduce **EAST**, an **E**fficient and **A**daptive **S**imultaneous machine **T**ranslation method with LLMs, which aims to achieve high-quality SiMT as efficiently as offline NMT. Specif-

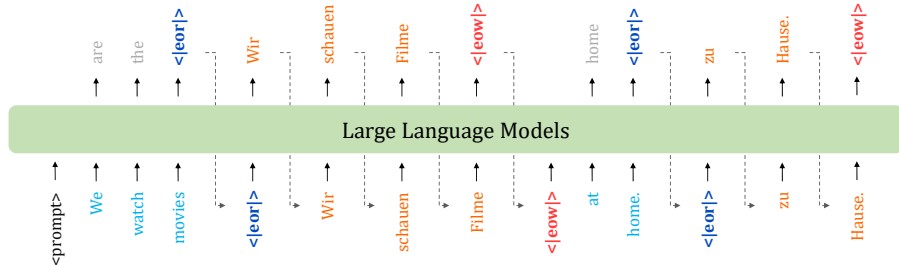

(a) Previous LLM SiMT with R/W policy, *e.g.*, wait-3: $p(y_{t-2}|\mathbf{x}_{\leq t}, \mathbf{y}_{\leq t-3})$, where KV cache needs to recalculate as $t$ increments.

(b) The inference process of EAST in auto-regressive manner $p(\mathbf{c}_t^y|\mathbf{c}_1^x, \mathbf{c}_1^y, ..., \mathbf{c}_t^x)$, where $\mathbf{c}_t^{\cdot}$ is the $t$-th chunk of source or target.

Figure 1: Comparison between existing LLM-based SiMT methods and our EAST. (a) Existing LLM-based SiMT typically reuses the sequence organization $\mathbf{x}_{\leq t}, \mathbf{y}_{\leq t'}$ in encoder-decoder framework, leading the prompt to generate new target tokens always changing. (b) Our inference follows the nature of auto-regressive decoding without recalculating the KV cache. For simplicity, we use `<|eor|>` and `<|eow|>` to represent the special tokens without hurting readability.

ically, we first leverage the instruction-following capability of LLMs to generate the SiMT data (Sakai et al., 2024) with different latency levels (low, medium, and high). Two SiMT datasets for supervised fine-tuning (SFT) are constructed, including the German-English dataset **SiMT-De-En-660K** and the multilingual dataset **SiMT-Multi-90K**. We structure the SFT data by alternating between the source and target segments of the generated SiMT data and introducing two special tokens (`<|end-of-read|>` and `<|end-of-write|>`) as explicit read-write signals. By performing SFT on this structured data on the LLMs, it can learn to effectively determine when to read more source input and when to generate the translation. A two-stage fine-tuning process is conducted to enhance its multilingual translation capabilities, with full-weight fine-tuning on the SiMT-De-En-660K dataset, followed by LoRA (Hu et al., 2022) fine-tuning on a combination of the MSiMT-90K and MNMT-120K datasets. During inference, EAST employs an adaptive read-write policy that aligns with its SFT recipe. The model predicts token-by-token and then dynamically switches between read and write actions based on whether the predicted token is a read or write signal. Due to the auto-regressive of our token I/O sequence, where new source input and target translations are incrementally appended, EAST can efficiently reuse the KV cache without modifying historical sequence. This significantly reduces computational costs and inference latency, improving the overall efficiency of SiMT. A comparison to highlight the main difference between previous LLM SiMT and ours is displayed in Figure 1. Experimental results show that EAST achieves high-quality SiMT across 8 translation directions and near-offline decoding speeds, without compromising offline translation performance, and generalizes well to document-level SiMT.

## 2 RELATED WORK

**Traditional SiMT** SiMT is a challenging task in machine translation, aiming to balance translation quality and latency. Unlike offline NMT, SiMT requires starting the translation before the full source sentence is available. The read-write policy governs this decision-making process by determining whether the system should wait to read more source text or proceed with writing the translation based

on the partial input. Traditional SiMT models are usually built on an encoder-decoder architecture or its variants, with fixed or trainable policies. A widely adopted fixed policy is the wait-$k$ (Ma et al., 2019), which has been extensively explored due to its simplicity (Elbayad et al., 2020; Zhang & Feng, 2021; Zhao et al., 2023; Zhang et al., 2023a; Fu et al., 2023; Wang et al., 2023b). However, it typically leads to suboptimal translations for complex contexts or non-monotonic language pairs (Zhang et al., 2022). Instead of relying on pre-defined rules, adaptive policies dynamically learn when to read and write, further improving the translation quality. To enable models to learn effective read-write decisions, a variety of techniques have been applied, including reinforcement learning (Gu et al., 2017; Arthur et al., 2021; Miao et al., 2021), dynamic programming (Miao et al., 2021; Liu et al., 2021), data augmentation Zhang et al. (2020); Deng et al. (2023), information transport theory (Zhang & Feng, 2022), and Hidden Markov Models (Zhang & Feng, 2023).

**LLM-based SiMT** The success of LLMs in offline NMT tasks has spurred interest in their application to SiMT. Recent research has explored various approaches to leverage LLMs for SiMT. However, the adaptive policies traditionally used in encoder-decoder architectures do not effectively apply to LLMs. Some methods (Wang et al., 2023a; Koshkin et al., 2024a;b; Agostinelli et al., 2024) have focused on optimizing prompts in conjunction with a fixed wait-$k$ policy. Moreover, Guo et al. (2024b) propose the Agent-SiMT framework, which incorporates a traditional adaptive SiMT model to assist in making read/write decisions. However, such methods face significant challenges, particularly as the need to update prompts during inference prevents the reuse of historical KV caches, resulting in inefficient recomputation and increased inference latency. The Agent-SiMT also complicates the process by requiring the training of an additional traditional SiMT model. Recently, some efforts (Sakai et al., 2024; Cheng et al., 2024) have utilized LLMs to generate SiMT data for learning adaptive policy, yet challenges persist with inefficiencies in LLM-based SiMT systems.

To improve translation efficiency, Wang et al. (2024) introduce the Conversational SimulMT framework, which employs a multi-turn dialogue decoding approach with generating SFT data by segmenting parallel sentences with an alignment tool. However, the tool can only output a fixed alignment configuration that is not easy for generalization. In addition, the method employs a fixed policy during inference that reads a fixed number of words at each step, leading to a mismatch with the fine-tuning process. Raffel et al. (2024) propose the SimulMask method to optimize the efficiency of the prompt-based approaches, by introducing a policy-specific attention mask during fine-tuning. This approach mimics the attention behavior during inference, restricting target token queries to only attend to the corresponding part of the source sequence in the prompt. However, SimulMask may not be suitable for adaptive policies because its complicated masking construction requires prior knowledge of the policy's decision process.

In addition, these SiMT methods primarily focus on leveraging the translation capabilities of LLMs, without exploring the adaptive read-write policies and generalization abilities. Unlike previous methods, EAST enables the LLMs to learn adaptive read-write policies in various latency requirements, and utilizes an interleaved text structure to significantly improve the efficiency of the LLMs during inference while maintaining consistency between fine-tuning and inference.

## 3 METHODS

There are three main challenges in applying LLMs to SiMT. **(i)** The updated prefix source and target sentences increase computational costs and latency due to KV cache recomputation. **(ii)** Adaptive read-write policies to support different latency requirements for LLM SiMT are still lacking, unless a fixed wait-$k$ policy is implemented. **(iii)** The generalization to multilingual language pairs may require an additional SFT data. To address these challenges, we propose EAST, an **E**fficient and **A**daptive **S**imultaneous machine **T**ranslation method with LLMs. The EAST approach involves three key components: the construction of SiMT data, the training of the LLMs on the SFT data, and the inference with adaptive read-write policy.

### 3.1 SiMT DATA CURATION VIA LATENCY-AWARE CHUNK SEGMENTATION

The availability of SiMT-specific datasets is scarce, and the annotation by professional interpreters is time-consuming and expensive. To address this issue, we leverage the powerful instruction-following capability of LLMs after RLHF (Ouyang et al., 2022) (*e.g.*, GPT-4 (OpenAI et al., 2024))

```
<|begin_of_text|><|start_header_id|>system<|end_header_id|>

You are a helpful assistant.<|eot_id|><|start_header_id|>user<|end_header_id|>

Translate the following text from English into German with low latency. <|eot_id|>
<|start_header_id|>assistant<|end_header_id|>

Anyone with information<|end-of-read|> Jeder, der Informationen hat,<|end-of-write|> is asked to call<|end-of-read|>
wird gebeten,<|end-of-write|> the SFPD Tip Line<|end-of-read|> das Hinweistelefon des SFPD<|end-of-write|> at 415-
575-4444.<|end-of-read|> unter 415-575-4444 anzurufen.<|end-of-write|><|eot_id|>
```

Figure 2: An example of the SiMT SFT data for Llama 3. Prompt is colored in gray. The source and target texts are highlighted in cyan and orange, respectively. The read-write tokens are highlighted in blue and red, respectively. We calculate the loss for all tokens other than the prompt during training.

and design a prompt that instructs LLMs to act as a professional simultaneous interpreter, segmenting sentences into independent semantic chunks and generating corresponding translations for each chunk.

In practice, SiMT must accommodate varying latency requirements depending on different use cases, such as live broadcasts that prioritize low latency and formal conferences that demand high-quality translation with higher latency. Importantly, different latencies naturally influence how sentences are segmented, reordered, and translated. Therefore, we prompt LLMs to generate SiMT data at three latency levels: "low", "medium", and "high". The prompt template is provided in Figure 12 in Appendix. Concretely, given a language pair $\mathbf{x}_{1:T_x}, \mathbf{y}_{1:T_y}$, the LLM output of the proposed low latency prompt can be represented as follows.

$$\mathbf{x}_{1:T_x} = [\mathbf{c}_1^x, \cdots, \mathbf{c}_{T_{low}}^x], \quad \mathbf{y}_{1:T_y} = [\mathbf{c}_1^y, \cdots, \mathbf{c}_{T_{low}}^y] \tag{1}$$

where $\mathbf{c}_t^{[\cdot]}$ is the $t$-th semantic chunk of source or target. With simple length filtering, the two chunk sequences should be well aligned with the same length[1]. Similarly, we can obtain the medium and high latency output. In general, for the same pair, we have $T_{low} \geq T_{medium} \geq T_{high}$.

In this study, we curated a dataset of 660K SiMT samples by extracting language pairs from the WMT15 De-En training data, allocating one-third of the samples to each latency requirement. While existing LLM-based SiMT methods typically train separate models for different language pairs (Guo et al., 2024b; Raffel et al., 2024), they often overlook the inherent multilingual capabilities of LLMs. In contrast, we constructed a smaller multilingual SiMT dataset of 90K samples encompassing eight translation directions.

### 3.2 TRAINING LLMs WITH SUPERVISED FINE-TUNING

To tackle the challenge proposed at the beginning of this section, we propose a two-stage SFT training process on the two curated SiMT datasets.

**Stage I: Activate SiMT of LLMs** The objective of this stage is to teach the LLMs how to perform adaptive simultaneous translation by learning when to read and write in our designed format. To enable the model to learn these adaptive behaviors, we reorganized the aligned chunks in the SiMT data by interleaving between source and target chunks and introduce two special tokens (`<|end-of-read|>` and `<|end-of-write|>`), *i.e.*,

$$[\mathbf{c}_1^x, <|eor|>, \mathbf{c}_1^y, <|eow|>, \cdots, \mathbf{c}_T^x, <|eor|>, \mathbf{c}_T^y, <|eow|>] \tag{2}$$

The special tokens act as explicit signals for the model to transition between reading and writing. The SiMT annotation process shows that each chunk contains enough semantic meaning for LLMs to carry out translations, ensuring that sequence reorganization does not lead to any loss of information for the model's reading or writing decisions. Since the annotation process also encodes the degree of fragmentation into latency indicator tokens—"low", "medium" or "high", the SFT can effectively guide the model to adapt to varying latency requirements. Figure 2 provides a comprehensive example of the SFT data.

---

[1] We first filter out source sentences with fewer than 20 words and then filter out examples where the number of source and target chunks generated by GPT-4 was not equal.

We train for one epoch during this stage on the larger SiMT-De-En-660K, employing full parameter tuning. Since SiMT defined in our proposed format is generally a novel task for LLMs, full parameter tuning ensures that the LLM can effectively and successfully learn the auto-regressive SiMT. Training for just one epoch helps mitigate the risk of overfitting during the full parameter tuning.

**Stage II: Generalize to Multilingual SiMT**  As the LLM acquires its auto-regressive SiMT capability in Stage I, its inherent multilingual proficiency enables it to generalize to multilingual SiMT, even with limited SFT data. Consequently, we apply LoRA (Hu et al., 2022) fine-tuning to a smaller multilingual dataset of 90K instances including eight language directions. Additionally, during this stage, we incorporate an offline NMT task to bolster the model's ability to translate full sentences and enhance overall translation performance. In fact, we can view offline translation as a specific instance of SiMT by treating the entire sentence as a complete semantic chunk, *e.g.*, $\mathbf{x}_{1:T} = \mathbf{c}_1^x$.

**Loss**  In previous works on the SiMT or LLM-based SiMT models, loss calculation *w.r.t.* source text is typically masked out, as it generally does not contribute to the training. However, in our case, the cross-entropy loss is calculated on both the target text and the source text, as well as on special tokens. The primary goal is to align with the auto-regressive design of interleaved sequences and establish the appropriate reading and writing timing.

### 3.3 Efficient Inference for Adaptive SiMT

During inference, EAST performs auto-regressive token-by-token prediction aligned with the training process as shown in Figure 1(b). The process unfolds in two main phases:

**Read-Predict-Discard**  During the read phase, the model sequentially receives source tokens and predicts the next token. If the predicted token is not `<|end-of-read|>`, it is discarded, and the next source token is appended to the current source chunk. Once `<|end-of-read|>` is predicted, the model transitions to the translating phase. Note that the discarding operation in the read phase does not violate the incremental appending of contexts, enabling EAST to efficiently utilize KV-cache for faster generation.

**Predict-Append**  Once the model enters the translation phase, it directly begins predicting the next token. If the predicted token is not `<|end-of-write|>`, it is appended to the current target chunk. When `<|end-of-write|>` is predicted, the model completes the current translation and returns to the reading phase.

Similar to the training phase, inference controls latency through indicator tokens—"low", "medium", or "high". Interestingly, an **Interpolation Effect** is observed, allowing for generalization to other latency levels using indicator tokens such as "low-medium" or "medium-high". Consequently, we can gather 3 to 5 observations to draw the BLEU-AL curve.

## 4 Experiments

### 4.1 Experimental Settings

**Datasets**  As introduced in the previous section, the primary SiMT SFT dataset to initiate novel task learning is **SiMT-De-En-660K**, derived from 660K parallel pairs from the WMT15 De-En training dataset. For each pair, we utilize LLMs to generate SiMT chunk sequences at three different latency levels, as outlined in Eq. (1), while filtering out invalid examples. In addition, we construct a smaller multilingual SiMT SFT dataset, **SiMT-Mult-90K**, which includes 8 language directions: De↔En, Zh↔En, Ru↔En, and Cs↔En. For offline NMT training, we collect datasets from WMT17 to WMT21, covering the same 8 translation directions, and refer to this collection as **Off-Multi-120K**. As shown in Table 4 of Appendix, the sentence-level **test data** is extracted from WMT22 across the same 8 translation directions as the offline NMT data. The majority of existing research primarily focuses on sentence-level evaluation. However, in many real-world applications, such as speech delivery, the input for SiMT often comes at the document level rather than isolated sentences. Moreover, LLMs have demonstrated impressive capabilities in long-form generation. Thus, we directly evaluate EAST on WMT22 *document-level* test data without additional fine-tuning.

**Evaluation Metrics** For quality evaluation, we use automatic metric–SacreBLEU[2] to compute the corpus-level BLEU, along with neural evaluation metrics BLEURT[3] (Sellam et al., 2020; Pu et al., 2021) and COMET[4] (Rei et al., 2020; 2022). For latency evaluation, we adopt Average Latency (AL) (Ma et al., 2019), computational-aware Average Latency (AL-CA)[5], and Length-Adaptive Average Lagging (LAAL) (Papi et al., 2022). In addition, we use Word Wall Time (WWT) Wang et al. (2024) to evaluate the model's decoding speed by calculating the actual inference time per word.

**Training Details** We train our models using Llama-3-8B-Instruct Dubey et al. (2024) as the backbone, with full parameter tuning on Stage I and LoRA tuning on Stage II. All models are trained on 8 Nvidia A100 GPUs with a total batch size of 256, a learning rate of 1e-5, a cosine learning rate scheduler, a warm-up ratio of 0.1 and a maximum sequence length of 1024. The number of epochs is set to 1 for full parameter tuning and 2 for LoRA tuning, respectively. When using LoRA, the LoRA rank, alpha, and dropout rate are set to 64, 128, and 0.05, respectively.

**System Settings** In this paper, we conduct comparative experiments between EAST and the following baselines. We employ the same training setup as EAST in these baselines.
(1) **EAST**: The proposed pipeline includes two-stage training, *i.e.*, full-weight fine-tuning on SiMT-De-En-660K followed by LoRA fine-tuning on SiMT-Mult-90K and Off-Multi-120K datasets.
(2) **EAST-Stage-I**: Full-weight fine-tuning on the SiMT-De-En-660K dataset.
(3) **EAST-Single-Stage**: Full-weight fine-tuning on all the three datasets for 1 single epoch.
(4) **EAST-w/o-Offline**: Removing the Off-Multi-120K datasets in Stage II.
(5) **EAST-Only-Stage-II**: Removing the Stage I fine-tuning.
(6) **Conversational SiMT** Wang et al. (2024): It is reproduced based on the Llama3 backbone model with EAST's SFT data and the two-stage training method for fair comparisons. During inference, it reads $k$ tokens each step and then incrementally decodes them.
(7) **Llama3-MNMT**: LoRA fine-tuning on the Off-Multi-120K dataset for offline NMT using Llama-3-8B-Instruct as the base model.
(8) **Llama3-NMT-De-En-660K w/ wait-$k$**: LoRA fine-tuning on the offline counterpart of SiMT-De-En-660K dataset dataset for offline NMT using Llama-3-8B-Instruct as the base model, and then applying the wait-$k$ policy for streaming inference.
(9) **Llama3-MNMT w/ wait-$k$**: Applying the wait-$k$ policy on the trained **Llama3-MNMT** model for streaming inference.

## 4.2 MAIN RESULTS

**SiMT X↔En** The BLEU-AL curves of SiMT X→En tasks are illustrated in first row of Figures 3. Notably, the EAST-Stage-I model, which is obtained from tuning the 660K De→En SiMT data alone, shows reasonable performance under varying latency instructions in language pairs like Ru→En and Cs→En due to linguistic similarities among these Indo-European languages, facilitating better transfer learning. In general, the EAST-Stage-I model exhibits superior performance over the Llama3-NMT-En-De-660K w/ wait-$k$ method for De→En, with +2 BLEU and +2 COMET at low latency (AL≤3) and +0.2 BLEU at high latency (AL≈8). This demonstrates the effectiveness of the SiMT-De-En-660K dataset tailored for learning the novel task. The Stage I model struggles with correct translation for Zh→En because of the significant structural and grammatical differences between Chinese and Indo-European languages. However, EAST with two-stage training greatly improves the performance of Zh→En becomes normal, underscoring the importance of this approach. Additionally, EAST with two-stage training further enhances performance across multiple latency ranges for De→En, Ru→En, and Cs→En, with improvements of 0.5 to 1 in BLEU.

The Stage I model completely fails to perform En→X translations, by repeating the source English sentence. This underperformance in En→X directions without specific fine-tuning on those language pairs highlights the challenges of cross-linguistic semantic structures in SiMT. Fortunately, a very smaller multilingual dataset with 90K SiMT parallel pairs enable the LLM excellent performance on the reversed language directions En→X. Compared with LLM-based SiMT baselines,

---

[2]https://github.com/mjpost/sacrebleu
[3]We use the recommended checkpoint BLEURT-20 for results reporting.
[4]https://huggingface.co/Unbabel/wmt22-comet-da
[5]The AL-CA metric is calculated by adding the machine processing time to the policy delay between the source text and target text.

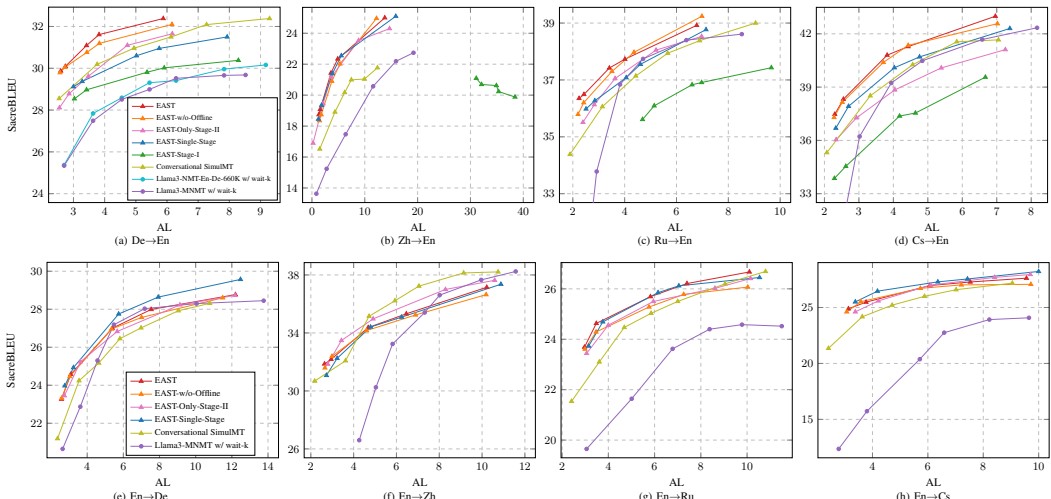

Figure 3: SacreBLEU against AL on the WMT22 X→En and En→X test sets.

EAST demonstrates superior performance across all 8 translation directions. In Figure 9 and 10 of the Appendix, we also plot the quality-latency curves of all methods with respect to COMET and BLEURT, revealing a trend similar to that of the BLEU-AL curves.

**SiMT Ablations** The EAST-w/o-Offline variant, which removes the offline NMT data in Stage II, shows a slight performance decline in De→En and Ru→En but maintained similar performance in other language pairs. The EAST-Only-Stage II that omits the Stage I fine-tuning results in a performance degradation of about 1 BLEU for the X→En on average, whereas the translation performance remains relatively unchanged for En→X. This suggests that learning a novel SiMT task may require a larger scale dataset. When fine-tuning with all three high-quality datasets in a single stage, EAST-Single-Stage demonstrates competitive performance across various delays and language orientations. Specifically, it achieves even higher BLEU-AL curves than the full two-stage training pipeline for both En→De and En→Cs. However, the two-stage training approach offers the advantage of better generalization to novel language directions with a reduced training schedule, avoiding re-training on the extensive Stage I dataset.

**Offline Performance** We also evaluate the performance of offline translation on the WMT22 test set, as presented in Table 1. Our results are superior to previous studies, Bayling (Zhang et al., 2023b) and ALMA (Xu et al., 2024a), except for a slight lag in En-Cs. Compared to offline NMT model Llama3-MNMT and other variants, EAST maintained comparable or superior offline translation performance across the eight language directions, indicating that our two-stage SFT process effectively maintains translation quality for offline NMT. EAST-Single-Stage shows excellent translation performance for En→X, although it slightly underperforms by about 3 BLEU and 1 COMET for X→En. The EAST-w/o-Offline model, not even trained on offline translation data, still performed well, particularly for X→En. This can be attributed to the fact that our high-latency SiMT data has context that is close to being as informative as the offline NMT data. Similar to the trend in SiMT, the offline performance of the EAST-Only-Stage-II drops by 1 BLEU and 0.22 COMET for X→En, while the performance remains relatively stable for En→X. In summary, these results highlight EAST not only excels in high-quality simultaneous translation but also ensures that the offline translation capabilities are not compromised.

### 4.3 ZERO-SHOT GENERALIZATION TO DOCUMENT-LEVEL SIMT

Since real-world applications often involve streaming inputs that are typically long and unsegmented, we further evaluate the EAST directly on the document-level test set from WMT22 De/Ru→En without fine-tuning on document-level data. In our experiments, the document-level test set is derived from the same data as the sentence-level set but without sentence segmentation.

| Models | De→En | | Zh→En | | Ru→En | | Cs→En | | Average | |
|---|---|---|---|---|---|---|---|---|---|---|
| | BLEU | COMET | BLEU | COMET | BLEU | COMET | BLEU | COMET | BLEU | COMET |
| GPT-4 | 33.87 | 85.62 | 27.20 | 82.79 | 43.51 | 86.18 | 48.67 | 87.43 | 38.31 | 85.51 |
| Bayling-7B | 28.16 | 83.19 | 20.31 | 77.48 | 34.74 | 82.48 | 35.98 | 82.03 | 29.80 | 81.30 |
| ALMA-7B-LoRA | 29.56 | 83.95 | 23.64 | 79.78 | 39.21 | 84.84 | 43.49 | 85.93 | 33.98 | 83.63 |
| Llama3-MNMT | 31.98 | **84.89** | **25.48** | **81.26** | 39.83 | **85.19** | 44.92 | **86.23** | **35.55** | **84.39** |
| EAST | **32.55** | 84.77 | 23.80 | 80.86 | 39.83 | 85.04 | **45.61** | 86.20 | 35.45 | 84.22 |
| EAST-Single-Stage | 30.01 | 84.15 | 24.05 | 80.20 | 36.06 | 84.39 | 39.12 | 84.63 | 32.31 | 83.34 |
| EAST-w/o-Offline | 32.37 | 84.55 | 22.42 | 80.85 | **40.29** | 84.80 | 41.21 | 85.41 | 34.07 | 83.90 |
| EAST-Only-Stage-II | 31.34 | 84.34 | 24.90 | 80.89 | 38.48 | 84.78 | 42.77 | 85.97 | 34.37 | 84.00 |

| Models | En→De | | En→Zh | | En→Ru | | En→Cs | | Average | |
|---|---|---|---|---|---|---|---|---|---|---|
| | BLEU | COMET | BLEU | COMET | BLEU | COMET | BLEU | COMET | BLEU | COMET |
| GPT-4 | 35.38 | 87.44 | 43.98 | 87.49 | 30.45 | 88.87 | 34.53 | 90.77 | 36.09 | 88.64 |
| Bayling-7B | 25.66 | 82.18 | 38.19 | 84.43 | 14.85 | 74.72 | 15.64 | 76.85 | 23.59 | 79.55 |
| ALMA-7B-LoRA | 30.16 | 85.45 | 36.47 | 84.87 | 26.93 | 87.05 | **30.17** | **89.05** | 30.93 | 86.61 |
| Llama3-MNMT | 30.45 | **85.63** | **40.68** | **86.53** | 24.83 | 87.27 | 27.92 | 88.36 | 30.97 | **86.95** |
| EAST | 30.84 | 85.49 | 40.17 | 86.31 | **26.79** | 87.13 | 26.63 | 88.17 | 31.11 | 86.78 |
| EAST-Single-Stage | **30.85** | 85.51 | 39.69 | 86.43 | 26.57 | 87.32 | 27.76 | 88.40 | **31.22** | 86.92 |
| EAST-w/o-Offline | 26.77 | 84.34 | 28.27 | 84.69 | 23.17 | 85.89 | 23.27 | 87.00 | 25.37 | 85.48 |
| EAST-Only-Stage-II | 30.50 | 85.44 | 39.03 | 86.31 | 26.62 | **87.40** | 26.85 | 88.32 | 30.75 | 86.87 |

Table 1: Offline results on the WMT22 X→En and En→X test sets. **Bold** values denote the highest scores, while the underlined values indicate the second highest scores for all models except GPT-4.

The results *w.r.t.* the corpus BLEU are depicted in Figure 4. EAST demonstrates superior performance in document-level settings, as this enhancement is due to the model's improved ability to leverage historical context, thereby enhancing translation accuracy and coherence.

Originally, document-level offline translation was expected to be one of the strongest capabilities of LLMs. Surprisingly, our proposed EAST model significantly outperforms both EAST and Llama3-MNMT in offline performance within a document-level context. This discrepancy in previous works may arise from models being trained exclusively on sentence-level data, which can lead to a training-inference mismatch during offline translation. Additionally, the long context of the source document may contribute to forgetting issues during the generation of the target document. However, our training approach, which alternates between source and target texts, effectively minimizes these mismatches. These results indicate that EAST is better suited for longer text sequences, making it particularly suitable for streaming scenarios.

Moreover, it can be observed that there is a significant rightward shift on the document-level BLEU-LAAL curve of the wait-$k$ method (Llama3-MNMT Doc-Si) compared to its sentence-level counterpart (Llama3-MNMT Sent-Si). Our statistical data indicates that English texts are substantially longer than their German and Russian counterparts in document-level test set—averaging 16.1 words longer than German and 35.8 words longer than Russian. This discrepancy is much greater than in the sentence-level test set, where English texts are only 2.2 and 3.1 words longer, respectively. Instead, EAST uses an adaptive read/write policy that effectively mitigates the above problems.

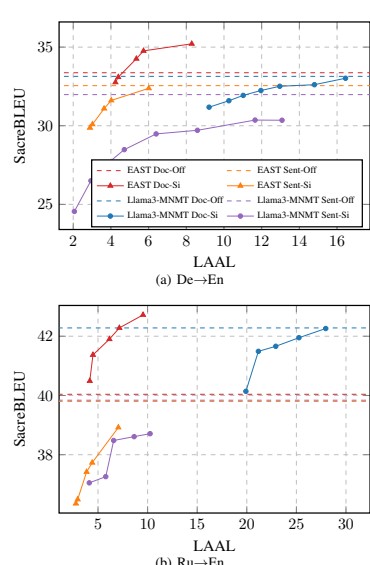

Figure 4: SacreBLEU-LAAL curves on the WMT22 document-level De/Ru→En test set. Methods labeled with "-Off" refer to offline translation, *i.e.*, including the entire document in the prompt. Methods marked with "-Si" denote simultaneous translation, involving the streaming input.

### 4.4 INFERENCE AS EFFICIENT AS OFFLINE

AL alone often fails to provide a comprehensive view of a translation system's efficiency, as it does not consider the computational costs involved in running the system. Thus, we measure the

| Method | BLEU (↑) | AL (↓) | AL-CA (↓) | WWT (ms) (↓) |
|---|---|---|---|---|
| EAST-offline | 32.55 | 14.62 | 15.22 | 38.96 |
| EAST | 29.87/31.08/32.38 | 2.59/3.42/5.87 | 3.26/4.29/6.55 | 49.87 (±1,21) |
| Llama3-MNMT w/ wait-k | 26.50/27.60/28.95 | 2.70/3.63/5.44 | 3.69/4.69/6.43 | 977.2 (±4.49) |

Table 2: Comparison of inference latency and speed on WMT22 De→En test set. The BLEU, AL, and AL-CA scores are given for low, medium, and high latency settings respectively. WWT refers to the actual inference time per word and is reported as mean and standard deviations (in parentheses) over the three latency.

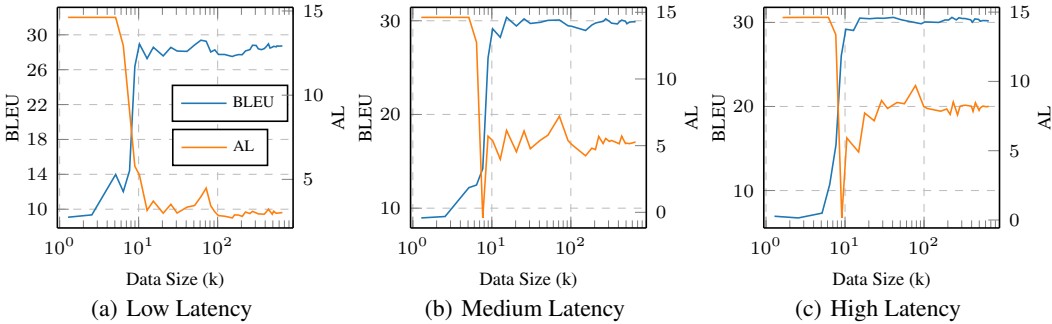

(a) Low Latency      (b) Medium Latency      (c) High Latency

Figure 5: BLEU scores (left $y$-axis ) and AL values (right $y$-axis ) over data size. We use log scale for scale the x-axis to more clearly observe the effect of data size. The original plots are also provided in Figure 11.

overall efficiency of the EAST model on an NVIDIA A100 through computational-aware latency (AL-CA) and decoding speed (WWT), as shown in Table 2. EAST achieves comparable translation performance to its offline counterpart EAST-offline with lower latency, and significantly outperforms Llama3-MNMT w/ wait-$k$ method under similar latency conditions.

For decoding speed, Llama3-MNMT w/ wait-$k$ shows the slowest inference speeds, taking up to 977.2ms to generate a single word. This inefficiency is attributed to the inability of this method to efficiently utilize the KV cache, necessitating the re-encoding of historical content at each decoding step, which limits its practical use in real-time scenarios. Conversely, EAST efficiently leverages KV-cache during inference, taking only 49ms to decode a word, achieving comparable decoding speeds to its offline counterpart (38.96ms per word). This small difference of about 10ms shows that EAST maintains near-offline efficiency, even under the streaming input conditions of SiMT.

## 4.5 DISCUSSIONS

**How many examples are needed to teach LLMs the novel SiMT task?** In this section, we investigate the data size required for efficiently training LLMs on the novel SiMT task based on the SiMT-De-En-660K dataset. Figure 5 illustrates the changes in BLEU scores and AL values with data sizes for different latency settings–"low", "medium", and "high". First, there is a significant improvement in BLEU score as the data size increases to approximately 10K. Beyond this point, the rate of increase in BLEU score diminishes. Similarly, AL metrics also decrease notably as the data size reaches around 10k, before stabilizing or showing minor fluctuations. This pattern is consistent across all latency settings, indicating a general learning behavior of the model: rapid enhancements in translation quality are achieved with the first 10K examples, followed by a phase where the model focuses on refining its read/write policy across different latency levels, up to 100k examples.

The results suggest that a relatively small dataset of just 10K SiMT examples may be sufficient to achieve commendable translation quality. This finding aligns well with our multilingual dataset, which contains approximately 10K examples per language, facilitating good performance across different languages. Moreover, expanding the data size (up to 100K examples) can further optimize the model's read/write policy.

**Can our approach replace traditional SiMT?** As Figure 6 shows, we compare our method with two categories of baselines: (1) Traditional SiMT methods, including ITST (Zhang & Feng, 2022), Mono-KD (Wang et al., 2023b), and SM$^2$-Bi (Yu et al., 2024); (2) LLM-based SiMT methods, including Agent-SiMT+HMT (Guo et al., 2024b) and C-SiMT (Conversational SimulMT Wang et al. (2024)). EAST achieves superior BLEU-AL curves, outperforming these traditional approaches with a large margin. We admit that the LLMs are typically pre-trained on extensive multilingual corpora, giving them an inherent advantage over smaller SiMT models. However, it is important to recognize that our definition of auto-regressive SiMT with an adaptive policy represents a completely novel challenge for LLMs, and the size of our SFT dataset considerably smaller than that of these methods. In addition, even the recent LLM-based SiMT C-SiMT and Agent-SiMT can only achieve on-par performance with traditional SiMT methods and does not show significant advantages.

To ensure a fair and thorough evaluation, we conduct additional experiments to isolate the contributions of our proposed dataset and adaptive strategy. When C-SiMT is trained on our SiMT-De-En-660K dataset, it achieves more than a 2 BLEU improvement across all latency settings compared to its original configuration using Llama2 and a larger dataset (4M examples). This demonstrates the effectiveness of our dataset. When incorporating our adaptive policy with Llama2 (EAST-Stage-I-Llama2), we observe an additional 0.9 BLEU improvement compared to C-SiMT-Llama2-SiMT-De-En-660K, showing the effectiveness of our policy. Upgrading the backbone model from Llama2 to Llama3 results in +0.5 BLEU in low-latency regions and +1 BLEU in high-latency regions.

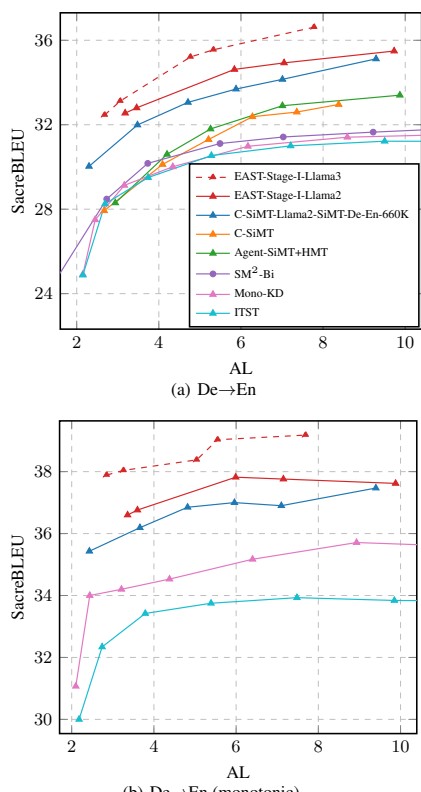

(a) De→En

(b) De→En (monotonic)

Figure 6: SacreBLEU against AL on the WMT15 De→En original test set and monotonic test set re-annotated by (Wang et al., 2023b).

Our approach addresses the first shortcoming of traditional SiMT—the lack of flexible read/write policies—while demonstrating the potential for even higher translation quality. The second shortcoming, related to computational cost, has been partially mitigated by our auto-regressive design. We believe that advancements in inference hardware will help close the speed gap between LLMs and smaller models.

## 5 CONCLUSION

In this paper, we introduce an **E**fficient and **A**daptive **S**imultaneous **T**ranslation method using LLMs, EAST, designed to achieve high-quality SiMT with the efficiency of offline systems. By constructing SFT data, leveraging an interleaved token structure with explicit read-write signals and incorporating latency-aware prompts, EAST enables LLMs to perform adaptive reading and translation based on varying latency requirements. Our experimental results demonstrate that EAST not only achieves state-of-the-art performance on SiMT benchmarks but also maintains high-quality translations in offline settings. Additionally, EAST shows excellent generalization to document-level SiMT, highlighting its suitability for streaming translation in real-world scenarios. The model's ability to reuse the KV cache during inference further ensures computational efficiency, allowing it to match the decoding speed of offline systems.

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

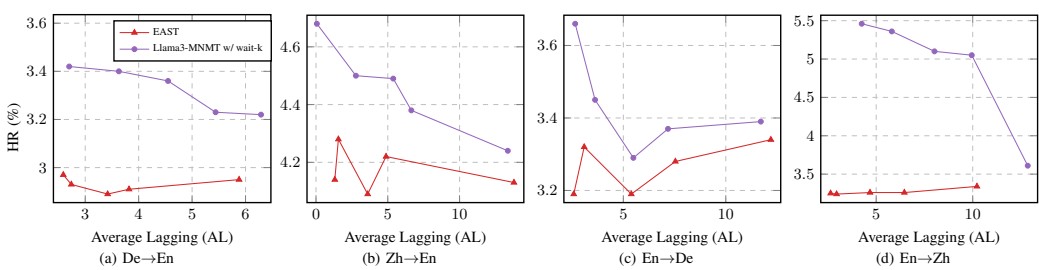

Figure 7: The hallucination rate (HR) against the latency metrics (AL) on the WMT22 test sets.

## A DATA STATISTICS

The data statistics for our SiMT datasets and offline NMT datasets are illustrated in Table 3 and 4, respectively.

| Latency | SiMT-Multi-90K | | | | | | | | | SiMT-De-En-660K |
|---|---|---|---|---|---|---|---|---|---|---|
| | De→En | Zh→En | Ru→En | Cs→En | En→De | EN→Zh | En→Ru | En→Cs | Total | De→En |
| Low | 3,325 | 1,635 | 3,642 | 2,507 | 3,267 | 2,423 | 4,945 | 4,226 | 25,970 | 230,902 |
| Medium | 3,631 | 2,763 | 3,719 | 2,472 | 3,997 | 2,433 | 5,830 | 5,035 | 29,880 | 227,131 |
| High | 4,102 | 4,166 | 4,254 | 2,746 | 4,921 | 2,920 | 6,322 | 5,433 | 34,864 | 202,843 |
| Total | 11,058 | 8,564 | 11,615 | 7,725 | 12,185 | 7,776 | 17,097 | 14,694 | 90,714 | 660,876 |

Table 3: The statistics for the two SiMT datasets we constructed.

| Language | Sentence-level Parallel Data | | | Document-level Parallel Data | | |
|---|---|---|---|---|---|---|
| | Train | Test (from En) | Test (to En) | Test (to English) | Avg. Words | Max. Words |
| German (De) | 14211 | 2037 | 1984 | 217 | 107/123 | 839/1003 |
| Chinese (Zh) | 15406 | 2037 | 1875 | - | - | - |
| Russia (Ru) | 15000 | 2037 | 2016 | 128 | 175/211 | 699/843 |
| Czech (Cs) | 12076 | 2037 | 1448 | - | - | - |

Table 4: The statistics for the parallel data from the WMT. "Avg. Words" indicates the average number of words per document in the source/target language. "Max. Words" represents the maximum number of words per document in the source/target language.

## B ADDITIONAL RESULTS

### B.1 WHAT IS HALLUCINATION RATE OF THE LLM-BASED SiMT?

Hallucination is a significant challenge in traditional SiMT, as the models begin translating while receiving input. This can prompt incorrect assumptions about the content yet to be received, resulting in hallucinated outputs. Additionally, hallucination is a common issue in the outputs of LLMs across various generation tasks. Therefore, it is more essential to evaluate the hallucination phenomenon in LLM-based SiMT. To effectively measure the hallucinations in our case, we utilize the hallucination rate (HR) metric (Chen et al., 2021), which quantifies the proportion of target words in the hypothesis that do not align with any source words. For this, we employ the fast-align[6] tool to identify word-level alignments between the source text and the target translation.

Figure 7 illustrates the HR comparison on En↔De and En↔Zh test sets. EAST consistently demonstrates a lower hallucination rate across all latency levels and test sets compared to the Llama3-MNMT w/ wait-$k$. Unlike the wait-$k$ policy, EAST can adaptively determine reading and writing actions based on the semantic context. This prevents the model from prematurely generating translations, thereby reducing the production of hallucinated content and ensuring translations that are more accurate and faithful to the source text.

---

[6]https://github.com/clab/fast_align

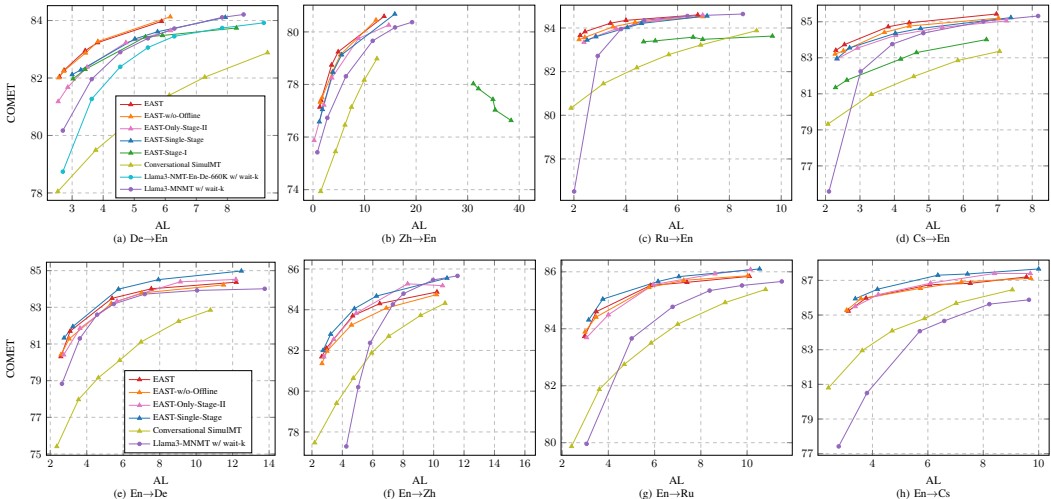

Figure 9: COMET against AL on the WMT22 X→En and En→X test sets.

## B.2 Quality of Translation Policy

To evaluate the quality of our translation policy, we conduct experiments on the manually aligned RWTH De→En alignment dataset[7]. Following (Zhang & Feng, 2022), we measure the proportion of ground-truth aligned source tokens that are read before generating each target token. Specifically, for a target token $y_i$, the number of source tokens read $(g_i)$ must be at least equal to the ground-truth aligned source position $(a_i)$. This ensures that the alignment between $y_i$ and $x_{a_i}$ is satisfied during the SiMT process. The proportion is calculated as follows:

$$A = \frac{1}{T} \sum_{i=1}^{T} \mathbb{I}_{a_i \leq g_i} \quad (3)$$

where $T$ is the total number of target tokens and $\mathbb{I}_{a_i \leq g_i}$ counts the number of $a_i \leq g_i$.

As shown in Figure 8, EAST consistently achieves the higher percentage of aligned source tokens read before translating across most latency levels compared to Conversational SimulMT and Wait-$k$. This result indicates that EAST better adheres to the ground-truth alignment, ensuring sufficient source context is read before generating target tokens.

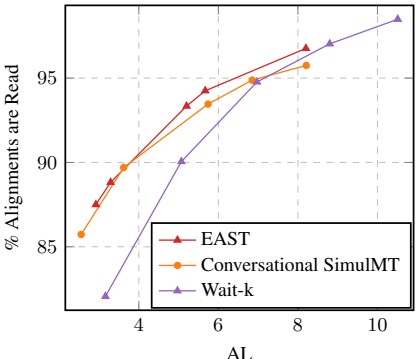

Figure 8: The proportion of the ground-truth aligned source tokens received before translating.

## B.3 Results of COMET and BLEURT

The COMET-AL and BLEURT-AL curves for sentence-level SiMT are provides in Figure 9 and 10. The normal scale counterpart of Figure 5 is illustrated in Figure 11.

## C Prompt

The prompt template for generating SiMT data is provided in Figure 12. The instruction data for offline translation is shown in the Figure 13.

---

[7]https://www-i6.informatik.rwth-aachen.de/goldAlignment/

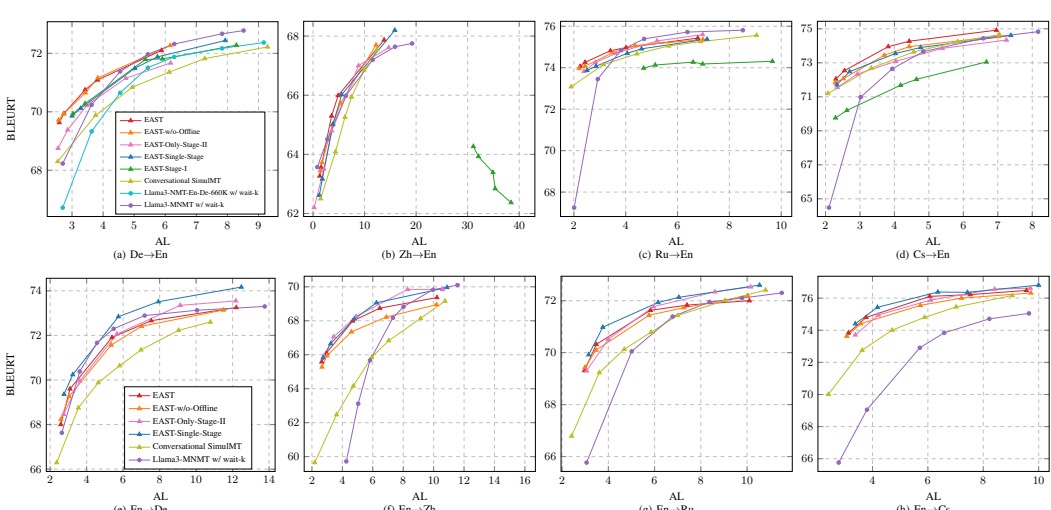

Figure 10: BLEURT against AL on the WMT22 X→En and En→X test sets.

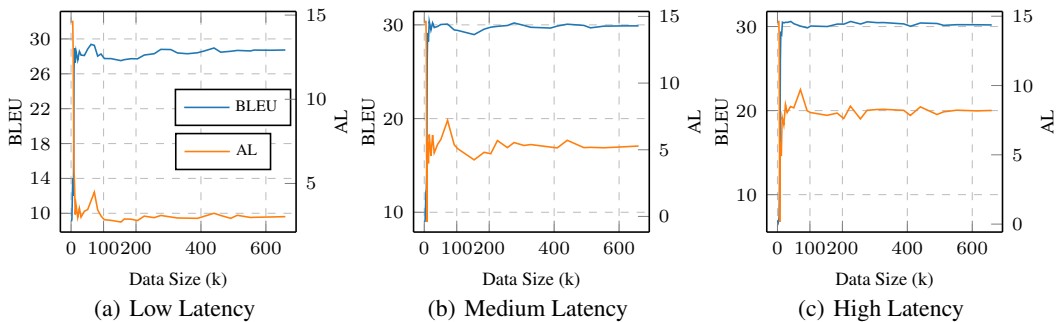

Figure 11: BLEU scores (left $y$-axis ) and AL values (right $y$-axis ) over data size.

As a professional simultaneous interpreter, your task is to segment sentences into independent semantic chunks and provide corresponding English translations.

You will use three different granularities for segmentation:
1. For low latency, the chunks would be fragmented into brief, coherent phrases that convey a complete thought.
2. For medium latency, the chunks would be longer, possibly clause or sentence-long segments.
3. For high latency, the chunks would be the longest, likely to cover complete clauses or full sentences.

You also need to provide corresponding simultaneous translation for each segment by performing the translation monotonically while making the translation grammatically tolerable.

Please take into consideration the example attached below:

Input:
Chinese: 休斯敦16日晚发出一系列龙卷风和严重雷暴警报。

Output:
```
{
    "low_latency": {
        "Chinese": ["休斯敦", "16日晚", "发出一系列", "龙卷风", "和严重雷暴", "警报。"],
        "English": ["Houston", "on the evening of the 16th", "issued a series of", "tornado", "and severe thunderstorm", "warnings."]
    },
    "medium_latency":{
        "Chinese": ["休斯敦16日晚", "发出一系列", "龙卷风和严重雷暴警报。"],
        "English": ["On the evening of the 16th, Houston", "issued a series of", "tornado and severe thunderstorm warnings."]
    },
    "high_latency": {
        "Chinese": ["休斯敦16日晚", "发出一系列龙卷风和严重雷暴警报。"],
        "English": ["On the evening of the 16th, Houston", "issued a series of tornado and severe thunderstorm warnings."]
    }
}
```

Figure 12: The prompt template for GPT-4 to generate SiMT data.

<|begin_of_text|><|start_header_id|>system<|end_header_id|>

You are a helpful assistant.<|eot_id|><|start_header_id|>user<|end_header_id|>

*Translate the following text from English into German.*

Anyone with information is asked to call the SFPD Tip Line at 415-575-4444. <|eot_id|>

<|start_header_id|>assistant<|end_header_id|>

Jeder, der Informationen hat wird gebeten das Hinweistelefon des SFPD unter 415-575-4444 anzurufen.

<|eot_id|>

Figure 13: An example of the offline NMT data for Llama 3. The source and target texts are highlighted in cyan and orange, respectively. We compute the loss on the target tokens during training.

