# OpenReview forum: "LLMs Can Achieve High-quality Simultaneous Machine Translation as Efficiently as Offline"
_ICLR.cc/2025/Conference — ICLR 2025 Conference Withdrawn Submission_

### Official Review · Reviewer_WkMG · 2024-10-17

**Soundness:** 1
**Presentation:** 3
**Contribution:** 2
**Rating:** 3
**Confidence:** 4

**Summary:**

The authors present a method for finetuning a large language model for simultaneous translation. They generate interleaved source-target data pairs by distilling a powerful LLM (i.e., GPT-4), which are then used to finetune a pretrained LLM. Experiments conducted on Llama-3-8B-Instruct demonstrate impressive numerical performance.

**Strengths:**

1. Experiments are conducted using state-of-the-art foundational LLM, Llama-3-8B-Instruct, demonstrating impressive numerical results.
2. Multilingual datasets and document-level datasets have been evaluated, which are often ignored in previous works.

**Weaknesses:**

1. Contribution is limited. The concept of using interleaved source-target text pairs for fine-tuning a large language model for simultaneous translation is not novel and has been discussed in prior work [1]. The paper's main contribution appears to be using a powerful LLM (i.e., GPT-4) to generate and rephrase this kind of data.
2. The use of LoRA to adapt a SiMT model for multilingual SiMT appears to be incremental and may not be highly relevant to the core focus of this work.
3. Unconvincing experiments. There is almost no comparison with recent LLM-based SiMT models (except Agent-SiMT) or state-of-the-art traditional SiMT. The comparison with Agent-SiMT is also not rigorous. **Since different foundation models are used, the advantages shown in your paper may stem from the foundational models or data distilled from GPT-4 (as the target text is modified!) rather than the simultaneous generation strategies.** A clear indication of this is that when AL is very large (i.e., close to offline translation), the translation performance gap between models widens, suggesting that streaming generation performance cannot be directly compared.


[1] Conversational SIMULMT: Efficient Simultaneous Translation with Large Language Models

**Questions:**

1. Clarify your contribution compared to paper [1]. **So far, the primary difference between the two works is that [1] employs an alignment-based method to create the interleaved dataset, while this work utilizes GPT-4 to generate and rephrase the data.**

2. Conduct rigorous comparative experiments! When comparing your methods with other LLM-SiMT models, at least ensure that the foundational models are consistent. If comparing with traditional SiMT models, you should maintain consistency in training data. According to the appendix (Figure 11), **using GPT-4 to create segmented texts in your method also involves *rephrasing* the target text. This can be seen as a form of distillation from GPT-4, which will definitely boost the performance for both your model and traditional SiMT models.**

---

> ### Author Response · Authors · 2024-11-23
> **Thanks for Reviewer WkMG's valuable comments!**
>
> We sincerely appreciate your time and effort in reviewing the paper and offering insightful feedback. We hope our responses below will clarify our intentions, address the key issues raised, and contribute to a re-evaluation of our work.
>
> **Q1:** Contribution is limited. The concept of using interleaved source-target text pairs for fine-tuning a large language model for simultaneous translation is not novel and has been discussed in prior work \[1\]. The paper's main contribution appears to be using a powerful LLM (i.e., GPT-4) to generate and rephrase this kind of data.
>
> **A1:** **See General Response I.**
>
> **Q2:** Conduct rigorous comparative experiments! When comparing your methods with other LLM-SiMT models, at least ensure that the foundational models are consistent. If comparing with traditional SiMT models, you should maintain consistency in training data.
>
> **A2:** **See General Response II.**
>
> **Q3:** According to the appendix (Figure 11), using GPT-4 to create segmented texts in your method also involves rephrasing the target text. This can be seen as a form of distillation from GPT-4, which will definitely boost the performance for both your model and traditional SiMT models.
>
> **A3:** We acknowledge that this can indeed be viewed as distillation from GPT-4. However, since GPT-4 is a closed-source system, it cannot be directly utilized to perform SiMT effectively in practice. Instead, we leverage GPT-4's powerful  instruction-following capabilities to construct high-quality SiMT datasets by segmenting and translating text. Our method then integrates these two capabilities into open-source LLMs, enabling them to perform SiMT efficiently and adaptively. This approach not only boosts performance but also extends the task boundaries of open-source LLMs, allowing them to handle the novel and challenging SiMT task.

---

> > ### Comment · Reviewer_WkMG · 2024-11-26
> >
> > I have carefully reviewed the authors' response. My major concerns remain unresolved.
> >
> > In their general response 1, the authors highlight that their contribution, beyond distilling knowledge from ChatGPT, lies in the use of different prompt templates compared to ConversationalSimulMT. However, these contributions seem incremental. As stated in their general response 2, most of the improvement comes from the use of distilled data and a stronger foundational model, which appears to be a trivial improvement.

---

### Official Review · Reviewer_pfiW · 2024-10-24

**Soundness:** 2
**Presentation:** 2
**Contribution:** 2
**Rating:** 3
**Confidence:** 4

**Summary:**

This paper introduces a unique dataset specifically designed for simultaneous machine translation, enabling large language models to perform this task. The paper primarily focuses on the methodology for constructing the simultaneous translation dataset. Experimental results demonstrate that the proposed dataset equips large language models with simultaneous translation capabilities and achieves expected performance in the experiments.

**Strengths:**

1. This paper proposes a dataset specifically designed for fine-tuning large language models for simultaneous machine translation.

2. Experimental results prove that the proposed dataset enables large language models to acquire a certain level of simultaneous translation capability.

**Weaknesses:**

1. The approach presented in this article is based on existing methods (i.e., Conversational simulmt) with some improvements, but its innovation is limited.
2. In the main experiment of Figure 3, EAST-Only-Stage-II outperforms EAST-Stage-I in the De-En direction, which is unexpected. This is because EAST-Only-Stage-II fine-tunes the LLMs using a multilingual dataset, while EAST-Stage-I fine-tuned the LLMs on a larger-scale De-En dataset. Can the author provide an explanation for this outcome?
3. In Line 298-300, this paper mention, "the EAST-Stage-I model exhibits superior performance over the Llama3-MNMT w/ wait-k method for De->En. This demonstrates the effectiveness of the SiMT-De-En-660K dataset tailored for learning the novel task". However, the comparison presented here does not sufficiently support the authors' claim. To better highlight the benefits of the constructed De-En SiMT dataset, the authors should fine-tune LLMs using an offline dataset, apply the wait-k strategy, and compare it with EAST-Stage-I.
4. In Section 4.4, the paper report the AL-CA metric, which is a latency metric for speech-to-text simultaneous translation. However, the paper does not specify which speech dataset was used for the experiment, nor does it explain how this metric was calculated.
5. In Figure 7, the authors compare their method with previous ones and claim significant improvement. However, their approach utilizes Llama3, while the previous methods use smaller Transformer models or Llama2. This comparison is not entirely fair and cannot showcase the effectiveness of the proposed method.
6. The focus of this paper is on using interleaved sequences to enable LLMs to generate next tokens based on the interleaved sequence in different languages. While this is a good starting point, we would like to see more analysis and comparison of the quality of the translation policies acquired by the fine-tuned LLMs, which is not presented in the paper.

**Questions:**

Pleased refer to the aboved weakness.

---

> ### Author Response · Authors · 2024-11-23
> **Thanks for Reviewer pfiW's valuable comments!**
>
> We sincerely appreciate your time and effort in reviewing the paper and offering insightful feedback. We hope our responses below will clarify our intentions, address the key issues raised, and contribute to a re-evaluation of our work.
>
> **Q1:** The approach presented in this article is based on existing methods (i.e., Conversational simulmt) with some improvements, but its innovation is limited.
>
> **A1:** **See General Response I.**
>
> **Q2:** In the main experiment of Figure 3, EAST-Only-Stage-II outperforms EAST-Stage-I in the De-En direction, which is unexpected. This is because EAST-Only-Stage-II fine-tunes the LLMs using a multilingual dataset, while EAST-Stage-I fine-tuned the LLMs on a larger-scale De-En dataset. Can the author provide an explanation for this outcome?
>
> **A2:** We appreciate the reviewer's insightful observation regarding the performance difference between EAST-Only-Stage-II and EAST-Stage-I in the De-En direction in Figure 3. We believe this outcome is primarily due to differences between the training and testing datasets.
>
> EAST-Stage-I is trained using the SiMT-De-En-660K dataset, which is derived from the WMT15 training set. The data distribution in SiMT-De-En-660K differs significantly from that of the WMT22 test set, leading to a performance gap when evaluated on WMT22. EAST-Only-Stage-II uses SiMT-Mult-90K and Off-Multi-120K, which are derived from WMT17 to WMT21 test sets. As a result, these datasets are closer in distribution to the WMT22 test set, providing EAST-Only-Stage-II with a better match to the evaluation data and yielding superior results on WMT22.
>
> When evaluating on the WMT15 test set, EAST-Stage-I consistently outperforms EAST-Only-Stage-II, as seen in the BLEU/AL scores in the table.
>
> We hope this explanation clarifies the observed results. Thank you for your insightful comment!
>
> |  Method  |  WMT15 |  |  |  WMT22 |  |  |
> | --- | --- | --- | --- | --- | --- | --- |
> |  EAST-Stage-I  |  **32.46/2.68**  |  **33.12/3.06**  |  **35.55/5.33**  |  28.54/3.03  |  28.97/3.42  |  30.02/5.91  |
> |  EAST-Only-Stage-II  |  30.06/2.53  |  31.22/3.18  |  33.03/5.73  |  **28.78/2.86**  |  **29.62/3.48**  |  **31.65/6.17**  |
>
> **Q3:** In Line 298-300, this paper mention, "the EAST-Stage-I model exhibits superior performance over the Llama3-MNMT w/ wait-k method for De->En. This demonstrates the effectiveness of the SiMT-De-En-660K dataset tailored for learning the novel task". However, the comparison presented here does not sufficiently support the authors' claim. To better highlight the benefits of the constructed De-En SiMT dataset, the authors should fine-tune LLMs using an offline dataset, apply the wait-k policy, and compare it with EAST-Stage-I.
>
> **A3:** We refine our experiments by fine-tuning LLMs using the offline counterpart of the SiMT-De-En-660K dataset and applying the wait-k policy for inference. The results of these additional experiments are now included in Figure 3 of the revised paper.
>
> **Q4:** In Section 4.4, the paper reports the AL-CA metric for speech-to-text simultaneous translation, but the paper does not specify which speech dataset was used or how the metric was calculated.
>
> **A4:** **See General Response III.**
>
> **Q5:** In Figure 7, the authors compare their method with previous ones and claim significant improvement. However, their approach utilizes Llama3, while previous methods use smaller Transformer models or Llama2. This comparison is not entirely fair.
>
> **A5:** **See General Response II.**
>
> **Q6:** The focus of this paper is on using interleaved sequences to enable LLMs to generate next tokens based on the interleaved sequence in different languages. We would like to see more analysis and comparison of the quality of the translation policies acquired by the fine-tuned LLMs.
>
> **A6:** Thank you for your valuable suggestion. We have addressed this by adding detailed results on the evaluation of translation policy quality to the appendix of the revised paper.

---

> > ### Comment · Reviewer_pfiW · 2024-11-23
> > **Response to authors**
> >
> > I appreciate the rebutal of authors. However, I will maintain my rating since the new results are not more convincing than the existing results.

---

### Official Review · Reviewer_7QfU · 2024-11-03

**Soundness:** 3
**Presentation:** 3
**Contribution:** 3
**Rating:** 6
**Confidence:** 5

**Summary:**

The paper presents EAST, an Efficient and Adaptive Simultaneous Machine Translation method using Large Language Models (LLMs) to improve real-time translation quality while achieving efficiency comparable to offline neural machine translation (NMT). EAST addresses challenges in existing SiMT systems by generating structured training data with explicit read-write signals and employing a dynamic read-write policy during inference that allows efficient reuse of the key-value (KV) cache. Experimental results indicate that EAST achieves high-quality translations across eight language pairs and near-offline decoding speeds, thereby enhancing the overall performance of simultaneous machine translation.

**Strengths:**

- Simultaneous machine translation is an interesting and promising research direction. The article is well written and easy to follow.
- EAST leverages the instruction-following capability of LLMs to create structured training datasets with explicit read-write signals, enabling effective learning of simultaneous translation strategies.
- Experimental results demonstrate that EAST achieves high-quality translations across eight language pairs while maintaining near-offline decoding speeds.

**Weaknesses:**

- The SiMT datasets used for training, SiMT-De-En-660K comes from WMT15 De-En. As far as I know, WMT15 De-En has about 4.5M parallel data. Why the author only rewrites 660K of the data, the motivation needs to be explained. If there is more data, will the performance be further improved?
- I look forward to an experiment on the performance trend with the amount of training data, which will help guide us to retrain LLM to perform SiMT with the minimum amount of data.
- In the main experimental results in Figure 3, it is suggested that some previous LLM-based SiMT methods can be added to demonstrate the effectiveness of EAST.

**Questions:**

Refer to weaknesses.

---

> ### Author Response · Authors · 2024-11-23
> **Thanks for Reviewer 7QfU's valuable comments!**
>
> We sincerely appreciate your time and effort in reviewing the paper and offering insightful feedback. We hope our responses below will clarify our intentions, address the key issues raised, and contribute to a re-evaluation of our work.
>
> **Q1:** The SiMT datasets used for training, SiMT-De-En-660K comes from WMT15 De-En. As far as I know, WMT15 De-En has about 4.5M parallel data. Why the author only rewrites 660K of the data, the motivation needs to be explained.
>
> **A1:** To ensure the highest possible quality of the SiMT data generated by GPT-4, we applied the following filtering steps:
>
> - We first filtered out source sentences with fewer than 20 words, as shorter sentences are less representative of real-world simultaneous translation scenarios and often do not require complex chunking strategies.
>
> - We further filtered out examples where the number of source and target chunks generated by GPT-4 was not equal.
>
> After applying these filters, we obtained a final dataset of 660K SiMT examples. We have revised manuscript clarifies this motivation and filtering process. Thank you for highlighting this point!
>
> **Q2:** If there is more data, will the performance be further improved? I look forward to an experiment on the performance trend with the amount of training data, which will help guide us to retrain LLM to perform SiMT with the minimum amount of data.
>
> **A2:**  In **Section 4.5**, we have already investigated the data size required for efficiently training LLMs on the SiMT task using the SiMT-De-En-660K dataset. The experimental results presented in Figures 5 and 10 indicate that a dataset of only 10K SiMT examples is sufficient to achieve commendable translation quality. Moreover, our findings show that increasing the dataset size beyond this threshold does not result in significant performance improvements.
>
> **Q3:** In the main experimental results in Figure 3, it is suggested that some previous LLM-based SiMT methods can be added to demonstrate the effectiveness of EAST.
>
> **A3:** Thank you for your suggestion to include more previous LLM-based SiMT methods in the main experimental results to better demonstrate the effectiveness of EAST.
>
> In the revised version of the paper, we have added the **Conversational SimulMT** baseline to Figures 3, 8 and 9. To ensure fair comparisons, we implemented it using the Llama3 backbone model, trained with EAST's SFT data, and applied the same two-stage training method as our approach. During inference, the Conversational SimulMT baseline adopts a policy of reading k tokens each step before incrementally decoding them.
>
> The experimental results in Figures 3, 8 and 9 show that, in terms of BLEU, EAST outperforms Conversational SimulMT in all directions except for En-Zh. Additionally, for BLEURT and COMET,  EAST surpasses Conversational SiMT by a large margin in all 8 translation directions.

---

### Official Review · Reviewer_o3Kw · 2024-11-04

**Soundness:** 3
**Presentation:** 3
**Contribution:** 2
**Rating:** 5
**Confidence:** 5

**Summary:**

This paper introduces EAST, a novel method for performing simultaneous machine translation with Large Language Models.  The authors address the limitations of LLMs in SiMT, such as the computational overhead of KV cache recomputation and the lack of adaptive read/write policies. EAST employs a two-stage supervised fine-tuning process. First, the authors resort to GPT to constructSiMT dataset  with interleaved source and target chunks marked by special read/write tokens which can be used to further finetune the model. Second, it fine-tunes the model on a multilingual SiMT dataset (SiMT-Multi-90K) and an offline MT dataset using LoRA.  The authors claim EAST achieves state-of-the-art performance on various benchmarks, maintains near-offline translation quality, and generalizes well to document-level SiMT without specific fine-tuning.

**Strengths:**

1. The use of an interleaved sequence with special read/write tokens for SFT is a novel approach that simplifies the training process and allows for efficient KV cache reuse during inference.
2. EAST learns an adaptive read/write policy without relying on traditional wait-k strategies, potentially leading to more nuanced translation decisions.

**Weaknesses:**

1. While the specific techniques are novel, the overall approach of fine-tuning LLMs for SiMT is not entirely new. The incremental contribution over existing LLM-based SiMT methods by prompting a LLM to construct a SiMT dataset to teach its format is not substantial enough to warrant acceptance.

2. The evaluation primarily focuses on BLEU, COMET, and BLEURT. More comprehensive evaluation metrics specifically designed for SiMT, such as latency-aware metrics and human evaluation, are necessary to validate the claims of high quality and efficiency. The "interpolation effect" claim lacks sufficient evidence and analysis

**Questions:**

1. The paper mentions an "interpolation effect" for latency.  Can you elaborate on this and provide more empirical evidence to support it?
2. How robust is EAST to noisy or non-fluent input, which is common in real-world streaming scenarios?
3. Do you consider converting the format learning for LLM into a RL approach?

---

> ### Author Response · Authors · 2024-11-23
> **Thanks for Reviewer o3Kw's valuable comments!**
>
> We sincerely appreciate your time and effort in reviewing the paper and offering insightful feedback. We hope our responses below will clarify our intentions, address the key issues raised, and contribute to a re-evaluation of our work.
>
> **Q1:** The evaluation primarily focuses on BLEU, COMET, and BLEURT. More comprehensive evaluation metrics specifically designed for SiMT, such as latency-aware metrics and human evaluation, are necessary to validate the claims of high quality and efficiency.
>
> **A1:** We would like to clarify that our paper already includes four **latency-aware metrics** to evaluate the performance of our method comprehensively:
>
> - **AL and LAAL**: Measure policy latency, quantifying the delay introduced by the translation policy.
>
> - **AL-CA**: Evaluates latency while incorporating machine processing time, offering a realistic assessment of system efficiency.
>
> - **WWT**: Assesses the decoding speed of the model, reflecting its computational efficiency.
>
> Furthermore, as shown in **Table 2**, our experimental results demonstrate that our method achieves high-quality translation while maintaining efficiency comparable to offline models. These latency-aware metrics, combined with translation quality metrics such as BLEU, COMET, and BLEURT to provide a comprehensive evaluation of the quality and latency for our method.
>
> **Q2:** The paper mentions an "interpolation effect" for latency. Can you elaborate on this and provide more empirical evidence to support it?
>
> **A2:** In our SFT data, we explicitly include three latency control tokens: low, medium, and high (e.g., low latency in Figure 2). These tokens are used during training to guide the model to produce translations at different latency levels. During inference, however, we can further introduce interpolated tokens, such as low-medium and medium-high, to achieve finer-grained control of latency. This interpolation enables us to adjust the latency beyond the three discrete levels seen during training. As a result, in EAST's AL-BLEU curve, we are able to plot five distinct points corresponding to the five latency levels (low, low-medium, medium, medium-high, and high). This demonstrates the model's ability to adapt to intermediate latency levels even if they were not explicitly defined in the training data.
>
> **Q3:** How robust is EAST to noisy or non-fluent input, which is common in real-world streaming scenarios?
>
> **A3:** Since our training data contains very limited noisy or non-fluent examples, EAST may not be inherently robust to such input in its current form. However, we believe that augmenting the training data with noisy or non-fluent samples could significantly enhance the model's robustness to such conditions. Training with noise-injected data has been shown in prior work to improve a model's ability to handle real-world variability effectively. It is an interesting direction for future research.
>
> **Q4:** Do you consider converting the format learning for LLM into a RL approach?
>
> **A4:** This is an interesting suggestion. Currently, we focus on SFT to learn the format for SiMT. We have not yet explored using RLfor this task, but we recognize the potential for RL to improve the adaptation of translation policies. It will be a great research direction for future work.

---

### Official Review · Reviewer_1UgB · 2024-11-05

**Soundness:** 2
**Presentation:** 2
**Contribution:** 1
**Rating:** 3
**Confidence:** 4

**Summary:**

Towards simultaneous machine translation, this paper mainly presents a novel dataset created for LLMs. The main contribution of the paper is the construction of the dataset for simultaneous translation. Compared to other methods, the proposed method enables LLMs to perform simultaneous translation effectively, achieving the desired performance in the experiments.

**Strengths:**

1. This paper provides a simultaneous translation dataset for the training of large language models (LLMs), enabling LLMs to perform simultaneous translation tasks in the form of multi-turn dialogues. This dataset is valuable for the training of similar simultaneous translation models in the future.

2. Subsequent experimental results show that the large language models fine-tuned with this dataset demonstrate the ability to perform simultaneous translation tasks.

**Weaknesses:**

1. Lack of Innovation. The innovation in this paper is limited. The idea of performing simultaneous translation tasks through multi-turn dialogues is relatively straightforward, and similar attempts, such as Conversational SimulMT[1], have already been made in the past. The main contribution of this paper lies in using GPT-4 to split a traditional full-sentence translation dataset into more granular dialogue-form data. While the authors also attempt to extend their method to multilingual simultaneous translation, this contribution is more of an engineering effort rather than a methodological innovation.

2. Unclear Experimental Description and Conceptual Errors. The analysis of the experiments is vague, with some fundamental misunderstandings. The paper reports the AL-CA metric in Table 2 but does not describe the speech translation dataset or provide information about the machine used for the experiments. Moreover, it is unclear how a text-based simultaneous translation model handles speech input. Therefore, regarding the AL-CA metric, which simultaneously accounts for waiting time of source input and machine processing time, the paper’s description and results are unsatisfactory.

3. Unfair Comparison. The paper compares its method to previous approaches such as Mono-KD, Agent-SiMT+HMT, and shows better results in Figure 7. However, the authors use Llama3 as the LLM for their method, while the other methods used Llama2 or smaller Transformer models. This comparison does not demonstrate the policy advantage of the proposed method, as the choice of a more powerful generation model already provides a significant advantage.

[1] Minghan Wang, Thuy-Trang Vu, Yuxia Wang, Ehsan Shareghi, Gholamreza Haffari. 2024. Conversational SimulMT: Efficient Simultaneous Translation with Large Language Models. In arXiv:2402.10552.

**Questions:**

The questions is shown in the Weakness.

---

> ### Author Response · Authors · 2024-11-23
> **Thanks for Reviewer 1UgB's valuable comments!**
>
> We sincerely appreciate your time and effort in reviewing the paper and offering insightful feedback. We hope our responses below will clarify our intentions, address the key issues raised, and contribute to a re-evaluation of our work.
>
> **Q1**: The innovation in this paper is limited.
>
> **A1**: **See General Response I.**
>
> **Q2**: The experimental description for the AL-CA metric is unclear.
>
> **A2**: **See General Response II.**
>
> **Q3**: The comparison between the proposed method and previous approaches is unfair.
>
> **A3**: **See General Response III.**

---

### Author Response · Authors · 2024-11-23
**General Response III to Reviewers' Comments on AL-CA Metric Description.**

We apologize for the unclear AL-CA description causing confusion to the reviewers. **The results in Table 2 were conducted on the WMT22 De-En test set, not on a speech dataset.**  While the AL-CA metric was initially introduced for simultaneous speech translation tasks, where it measures latency by adding machine processing time to the policy delay between the source speech and target text, we adapted it to the SiMT task by applying a similar principle. In our work, the AL-CA metric is calculated by adding the machine processing time to the policy delay between the source text and target text.

Thanks for pointing this out, and we have revised the paper to include a more detailed explanation.

---

### Author Response · Authors · 2024-11-23
**General Response II to Reviewers' Comments on Fair Comparison.**

Thanks for raising this concern, and we have revised the manuscript to provide a clearer explanation of this point.

As shown in Figure 6, we compare our method with two categories of baselines: (1) Traditional SiMT methods, including ITST, Mono-KD, and SM2-Bi; (2) LLM-based SiMT methods, including Agent-SiMT+HMT and C-SiMT (Conversational SimulMT). Compared to traditional SiMT methods, EAST-Stage-I achieves superior BLEU-AL curves, outperforming these traditional approaches with a large margin. In addition, the recent LLM-based SiMT C-SiMT and Agent-SiMT only achieve on-par performance with traditional SiMT methods and does not show significant advantages.

To ensure a fair and thorough evaluation, we conduct additional experiments to isolate the contributions of our proposed dataset and adaptive policy. When C-SiMT is trained on our SiMT-De-En-660K dataset, it achieves more than a 2 BLEU improvement across all latency settings compared to its original configuration using Llama2 and a larger dataset (4M examples). This demonstrates the effectiveness of our dataset. When incorporating our adaptive policy with Llama2 (EAST-Stage-I-Llama2), we observed an additional 0.9 BLEU improvement compared to C-SiMT w/ Llama2 and SiMT-De-En-660K, demonstrating the effectiveness of our policy. Upgrading the backbone model from Llama2 to Llama3 results in +0.5 BLEU in low-latency regions and +1 BLEU in high-latency regions.

We hope this clarification demonstrates that the effectiveness of our proposed method is not solely reliant on the choice of a stronger LLM. Instead, our contributions—including the dataset, adaptive policy, and overall approach—offer substantial improvements, independent of the backbone model.

---

### Author Response · Authors · 2024-11-23
**General Response I to Reviewers' Comments on Innovation. (Part 2/2)**

### Contributions of Our Work
To further clarify our methodological innovations and address concerns, we highlight the contributions of our work:
- We construct two novel latency-aware datasets, including a German-English dataset (SiMT-De-En-660K) and a multilingual dataset (SiMT-Multi-90K). Latency-awareness, a often neglected factor in prior SiMT studies, is carefully considered. As noted by reviewers, these datasets are valuable resources for future research in SimulMT.
- We propose a novel LLM-based adaptive read/write policy which achieves high-quality simultaneous translation as efficiently as offline model. To the best of our knowledge, this is the first efficient and adaptive LLM-based SiMT method.
- Experimental results on multilingual and document-level datasets demonstrate the effectiveness of our method, where document-level evaluation is particularly underexplored in prior work.
- We investigate the data size required for efficiently training LLMs on the novel SiMT task. Our findings reveal that only 10K SiMT examples may be sufficient to achieve commendable translation quality, offering valuable insights for future research.

We hope this clarification addresses the reviewers' concerns about the innovation of our work.

---

### Author Response · Authors · 2024-11-23
**General Response I to Reviewers' Comments on Innovation. (Part 1/2)**

We appreciate the reviewers' thoughtful comments and acknowledge that Conversational SimulMT and our method both adopt an interleaved source/target data structure. However, it's important to note that the contribution of a manuscript cannot be judged solely by the data format employed. Moreover, the detailed data structures differ, serving distinct functionalities in each approach. To address the concerns about the novelty of our approach, we would like to highlight the significant differences between Conversational SimulMT and our method, EAST, as well as clarify the contributions of our work.

### Key Differences from Conversational SimulMT
#### **Data Construction**
Conversational SimulMT relies on an alignment tool and multi-step data augmentation to create SiMT data. However, this approach has significant limitations:
- The resulting subsequences may not represent meaningful semantic units.
- The data construction method is based on offline parallel data, which introduces a domain mismatch with the SiMT paradigm.
- The word alignments generated by the alignment tool fast_align used in Conversational SimulMT have an error rate of approximately 30%, potentially degrading the quality of synthetic data and leading to suboptimal performance.

In contrast, EAST leverages GPT-4 to segment source text into independent semantic units and generate corresponding simultaneous translations, effectively mitigating these issues. Moreover, our data construction considers varying latency levels, as the models should generate different translations for different latency requirements, an important consideration often overlooked in prior works.
As shown in Figure 6 of the revised paper, when training our SiMT-De-En-660K dataset in Conversational SimulMT format, we observe a consistent performance improvement of 1 BLEU across all delay settings. Notably, our dataset (660K samples) is an order of magnitude smaller than that of Conversational SimulMT (4M samples). Moreover, the results in Figure 5 show that only 10K SiMT examples may be sufficient to achieve commendable translation quality.

**This concludes that a limited amount of high-quality, latency-aware aligned data is more effective than large-scale, tool-aligned data.**

#### **Training for Adaptive Read/Write Policy**
In Conversational SimulMT, the SFT data is structured into a chat (or message API) format during training, and only the loss on target tokens is computed to optimize the LLM's translation ability for incomplete text. However, during inference, it adopts a fixed policy (e.g., reading a fixed number of tokens at each step), leading to a mismatch between training and inference phases.
In contrast, EAST introduces the training method the same as the pretraining, i.e., or next-token prediction (or completion API) format, to learn an adaptive read/write policy. Particularly, the loss is  computed across source, target, and read/write tokens. This not only optimizes the translation performance but also enables LLMs to model adaptive read/write behaviors based on context, ensuring consistency between training and inference. As a result, EAST can effectively segment and translate source text into appropriate semantic units based on latency requirements specified by the instructions, achieving a latency-specific adaptive read/write policy.

#### **Computational Cost**
While the multi-turn dialogue format used in Conversational SimulMT can improve inference efficiency through KV cache reuse, it introduces extra role-specific delimiting tokens (e.g., for Llama3, one dialogue round adds 8 delimiting tokens including `<|start_header_id|>user<|end_header_id|>\n\n` and `<|start_header_id|>assistant<|end_header_id|>\n\n`).  These tokens significantly expand sequence length, increasing computational cost during fine-tuning and inference, especially in streaming scenarios where the source text is relatively long.  Our method avoids such structural overhead, as we do not format data into a standard multi-round dialogue format, resulting in a more efficient approach.

---

### Note · Authors · 2024-12-16

I have read and agree with the venue's withdrawal policy on behalf of myself and my co-authors.